# Safety and Impact of Anti-COVID-19 Vaccines in Psoriatic Patients Treated with Biologics: A Real Life Experience

**DOI:** 10.3390/jcm10153355

**Published:** 2021-07-29

**Authors:** Nevena Skroza, Nicoletta Bernardini, Ersilia Tolino, Ilaria Proietti, Alessandra Mambrin, Anna Marchesiello, Federica Marraffa, Giovanni Rossi, Salvatore Volpe, Concetta Potenza

**Affiliations:** Dermatology Unit “D. Innocenzi”, Sapienza University of Rome, Polo Pontino, 04100 Terracina, Italy; nicoletta.bernardini@libero.it (N.B.); ersilia.tolino@gmail.com (E.T.); proiettilaria@gmail.com (I.P.); mambrinalessandra@gmail.com (A.M.); anna.marchesiello90@gmail.com (A.M.); fe.marraffa@gmail.com (F.M.); rossi.1598580@studenti.uniroma1.it (G.R.); sal.volpe@hotmail.it (S.V.); concetta.potenza@uniroma1.it (C.P.)

**Keywords:** COVID-19, vaccine, psoriasis, biologic therapy

## Abstract

Since all clinical trials conducted during the development of anti-COVID-19 vaccines have adopted among the exclusion criteria the presence of immunodepression or immunomodulating therapy, to date, the effects of vaccination against the new coronavirus 2 in people under such conditions have yet to be clearly defined. The primary objective of the study is to assess the safety of treatment with biotechnological drugs in patients suffering from moderate–severe psoriasis and subjected to the prophylactic vaccination against SARS-Cov-2. Additionally, the secondary objective of the research is to investigate the existence of a possible impact of anti-COVID-19 vaccination on the natural chronic-relapsing course and the severity of the psoriatic disease. The study included 436 patients with moderate–severe psoriasis, both male and female, in treatment with biologics. The data were collected using the direct interview method. A reduction of 74.13% of average Psoriasis Area Severity Index (PASI )compared to baseline (T0) was found in all subjects; this does not differ significantly from the group that underwent vaccination (73.4%). Moreover; at the end of the study, neither mild nor severe adverse events (ADR) were observed among them. In conclusion, biotechnological drugs used in the management of patients with moderate–severe psoriasis demonstrate a high safety profile also in subjects immunized against SARS-Cov-2.

## 1. Introduction

Since all clinical trials conducted during the development of anti-COVID-19 vaccines have adopted among the exclusion criteria the presence of immunodepression or immunomodulating therapy in course, to date, the effects of these vaccines in such patients have yet to be clearly defined [1,2].

Currently, it seems that there is no evidence of a possible influence of vaccines against SARS-Cov-2 on the course and severity of psoriatic disease. Therefore, the International Psoriasis Council (IPC) recommends patients with psoriasis, who do not have specific contraindications, to receive one of the vaccines approved for the prevention of COVID-19 as soon as possible [3].

In addition to this recommendation, the National Psoriasis Foundation (NP) suggests that systemic and biological treatments should be maintained in patients who undergo vaccination. This advice is based essentially on the lack of guidance on the suspension of the same therapies and on the assessment of the risks and benefits that could result from their eventual interruption [4].

The primary objective of the study was to assess the safety of treatment with biotechnological drugs in patients suffering from moderate–severe psoriasis and subjected to the prophylactic vaccination against SARS-Cov-2. Although the high safety profile of innovative biological agents has been demonstrated by several studies, currently there is not enough pharmacovigilance data to define the level of safety of these drugs in psoriatic subjects who have received one of the anti-COVID-19 vaccines. The secondary aim was to investigate if there is the impact of anti-COVID-19 vaccination on the natural chronic-relapsing course and the severity of psoriatic disease.

## 2. Materials and Methods

### 2.1. Study Population and Design

The observational prospective monocentric real-life study was performed on 436 psoriatic patients from outpatient services. The study included subjects, both males and females aged over 18, in therapy with biological drugs (anti-TNF-α, anti-IL-17, anti-IL-12/23, anti-IL-23, anti-PDE4) for at least 24 weeks. From 436 patients affected by moderate to severe psoriasis, 78 underwent anti-COVID mRNA vaccination in the meantime.

### 2.2. Data Collection

The variables were collected using the direct interview method, during the medical consultation that took place from January to March 2021.

The following parameters were reported at the baseline, time T0: -Sex, age, weight, and height;-Employment;-History of significant co-morbidities;-Type of biological drug prescribed;-The initial value of the Psoriasis Area and Severity Index (PASI).

After six months from the beginning of the protocol, at time T1, the patients admitted to the research were re-evaluated; in particular, the information collected were:-Anti-COVID-19 vaccination in meantime;-Occurrence of serious adverse reactions (ADR) due to vaccination;-Exacerbation or clinical worsening of psoriatic disease, following vaccination;-Final evaluation of the PASI.

## 3. Results

From 436 subjects included in the study, 78 underwent anti-COVID19 mRNA vaccination. The study population was represented as follows: 263 males (60.32% of the total) and 173 females (39.68% of the total); the average age at the time of recruitment was 57.26 years. The mean value of the body mass index (BMI) was 24.82 Kg/m^2^. However, 187 patients (42.89%) presented one or more significant comorbidities.

Overall, 23.39% underwent therapy with an inhibitor of tumor necrosis factor (TNF-α), 43.35% with an interleukin 17 (IL-17) inhibitor, 13.30% with an antagonist of IL-12 and IL-23, 22.67% with anti-IL-23, and 10.09% with anti-PDE4. 

The average value of the initial PASI at baseline was 16.95 (T0).

However, 78 patients had received the mRNA vaccine of Pfizer/Biontech in the meantime because they were health-workers. Moreover, almost all of the vaccinated patients (97.44%) had received both doses necessary to induce an adequate immune response.

At the end of the study, it was found that none of the anti-COVID-19 vaccinated patients in concomitant therapy with anti-psoriatic immunomodulating agents developed adverse events (ADR), neither mild nor severe.

The average value of the final PASI, assessed at the end of the study (T1) was 3.01.

At the end of the study, a clinical improvement with PASI reduction of 74.13% from baseline (T0) was observed (Figure 1a). Similarly, in the selected group of the 78 patients who received anti-COVID-19 vaccine, the PASI reduction was 73.4%. (Figure 1b).

## 4. Discussion

This study was carried out between January and March 2021, the period in which the anti-COVID-19 vaccination campaign had started in Italy, with mandatory vaccination of healthcare workers [5]. In fact, the 78 psoriatic patients undergoing therapy with biological drugs, who had taken part in the anti-COVID-19 vaccination programme, reported to be health-workers. In addition, patients had completed vaccination with the Pfizer/Biontech mRNA vaccine. In fact, according to guidelines provided by the NPF Task Force, all patients with psoriatic disease should receive a COVID-19 mRNA vaccine. Furthermore, the authors point out that biological drugs for psoriasis or PsA are not in contraindication with the COVID-19 mRNA vaccine [6].

The same task force recommended that patients who were to receive a COVID-19 mRNA vaccine should not discontinue their biological therapies [6].

In this survey, data collection took place before the final announcement of these NPF guidelines. Therefore, the 78 vaccinated patients did not discontinue their current biological therapy, but for greater safety, they were instead advised to postpone its administration for 10 days after vaccination. However, the mRNA anti-COVID-19 vaccination in the study population has proven to be safe, causing no significant side effects.

At the end of the study, a clinical improvement with PASI reduction of 74.13% from baseline (T0) was observed. Analogously, in the group who received anti-COVID-19 vaccines, the PASI reduction was 73.4%.

This slight non-statistically significant difference in the group of vaccinated patients could be explained by the delay in the scheduled administration of the biological drug.

## 5. Conclusions

The immune response induced following administration of the anti-COVID-19 vaccine is not only humoral (with the production of neutralizing antibodies directed against the S protein of the viral capsid), but also cellular (with the activation and proliferation of various lymphocyte populations, in particular CD 8 + cytotoxic T lymphocytes, capable of rapidly detecting and eliminating infected cells) [7,8,9].

From this premise, the hypothesis of possible reactivation of the psoriatic skin lesions, as a consequence of vaccination, could be advanced. However, such side effects did not emerge from this real-world observational study.

Limitations of the study include the number of vaccinated patients (78), which might be too low to detect rare adverse events, and the single center design. However, our results suggest that anti-COVID-19 mRNA vaccines do not lead to a change in clinical response to biological treatments in the specific population studied.

Moreover, real-life data on the effect of COVID-19 mRNA vaccination in biologically treated subjects are currently lacking. Therefore, clinical experience in this population is needed in terms of safety and drug interactions.

In this real-life study, a 73.4% reduction in mean PASI was found among vaccinated subjects compared to the initial value; this reduction was not significantly different from that recorded in the total sample population, which was 74.13%.

Our results suggest that anti-COVID-19 mRNA vaccines do not lead to a change in clinical response to biological treatments in psoriatic patients. Moreover, this vaccine demonstrated its safety, so it could be recommended also in this fragile population [6].

The results of this study may be a further contribution in favor of anti-COVID-19 vaccination management in psoriatic patients.

## Figures and Tables

**Figure 1 jcm-10-03355-f001:**
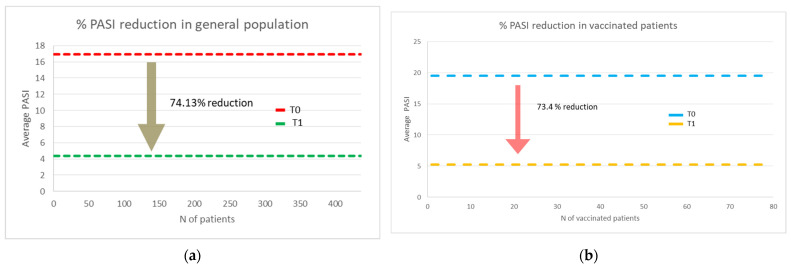
Average PASI at baseline and after 24 weeks in (**a**) all the vaccinated and non-vaccinated subjects, and (**b**) in vaccinated subjects.

## Data Availability

The data that support the findings of this study are available from the corresponding author upon reasonable request.

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
