# Peer review of "Safety and Impact of Anti-COVID-19 Vaccines in Psoriatic Patients Treated with Biologics: A Real Life Experience"

_jcm, 2021, doi:10.3390/jcm10153355_

Round 1

Reviewer 1 Report

The aim of the study is interesting, but the power to detect any difference is low and the paper structure and writing should be improved.

The number of vaccinated patients is not clearly stated at the beginning of the results ( 78?).

The group is highly selected and this is likely to bias the results and hamper external validity.

The primary outcome is the safety of vaccines. The study is clearly underpowered for this aim, and in fact, it is not discussed in the results or discussion. Most results are focused on the secondary outcome (effect of vaccination on drug effectiveness)

The first paragraph of the conclusion discusses mechanisms of disease and does not come as a consequence of the findings of the paper.

Author Response

Point 1: The aim of the study is interesting, but the power to detect any difference is low and the paper structure and writing should be improved.

Response 1: We modified the original draft in several parts. We substituted the section “2.1 Study population and design” with the following: “The observational prospective monocentric real-life study on 436 psoriatic patients from outpatient service was performed. The study included subjects, both males and females aged over 18, in therapy with biological drug (anti-TNF-α, anti-IL-17, anti-IL-12/23, anti-IL-23, anti-PDE4) for at least 24 weeks. From 436 patients affected by moderate to severe psoriasis, 78 underwent anti-COVID mRNA vaccination in meantime” (line 61-65). Minor reviews were taken in the sparagraph “Conclusion”: we added “drug interations” (line 140)  and changed “this vaccines with a generally demonstrated safety, could be recommended also in this fragile population” with  this vaccine generally demonstrated its safety, so it could be recommended also in this fragile population” The other changes are mentioned below. The other changes are mentioned below.

Point 2: The number of vaccinated patients is not clearly stated at the beginning of the results ( 78?).

Response 2: We added the sentence “From 436 subjects included in the study, 78 underwent anti-COVID19 mRNA vaccination” (line 82-83)

Point 3: The primary outcome is the safety of vaccines. The study is clearly underpowered for this aim, and in fact, it is not discussed in the results or discussion. Most results are focused on the secondary outcome (effect of vaccination on drug effectiveness)

Response 3: We added “However, the mRNA anti COVID vaccination in study population, has proven to be safe, causing no significant side effects” (line 122-123)

Point 4: The first paragraph of the conclusion discusses mechanisms of disease and does not come as a consequence of the findings of the paper

Response 4: We substituted the sentence: “It could be, thus, postulated that the reaction of the immune system following vaccination may significantly influence the course of the disease, with the possible reactivation of the skin lesions typical of psoriasis” with the sentence “From this premise, the hypothesis of a possible reactivation of the psoriatic skin lesions, as a consequence of vaccination, could be advanced.

However, such side effect did not emerge from this real-world observational study”.

Reviewer 2 Report

This concise clinical report has an essential value for  the patients suffered from Psoriasis treated immunomodulating therapy to show whether anti-COVID-19 vaccination is safe or need some attention. The paper is well organized and aim, method, introduction is fine, and the results is clear.

There are some questions to improve this important clinical manuscript further.

1. In Figure, what does the word “general population” mean in legend ?  In other words, please provide %PASI reduction value in non-vaccinated Psoriatic patients.

2. Are there any significant differences in %PASI among different therapies between vaccinated and non-vaccinated patients?

3. Where is Figure 1 ?  Also no description regarding Figure 2 in the main text.

Author Response

Point 1: This concise clinical report has an essential value for  the patients suffered from Psoriasis treated immunomodulating therapy to show whether anti-COVID-19 vaccination is safe or need some attention. The paper is well organized and aim, method, introduction is fine, and the results is clear.

There are some questions to improve this important clinical manuscript further.

In Figure, what does the word “general population” mean in legend ? 

Response 1: In the figure legend, we substituted “general population” with “all the vaccinated and not vaccinated subjects”

Point 2: Are there any significant differences in % PASI among different therapies between vaccinated and non-vaccinated patients? In other words, please provide % PASI reduction value in non-vaccinated Psoriatic patients.

Response 2: The % PASI reduction value in non-vaccinated Psoriatic patients was 74,28 %.

Point 3: Where is Figure 1 ?  Also no description regarding Figure 2 in the main text.

Response 3: We added “(Figure 1a)” and “(Figure 1b)” in the main text (line 97-98) and changed in the figure legend “Figure 2” with Figure 1.
